# Interplay between Residual Protease Activity in Commercial Lactases and the Subsequent Digestibility of β-Casein in a Model System

**DOI:** 10.3390/molecules24162876

**Published:** 2019-08-08

**Authors:** Di Zhao, Thao T. Le, Lotte Bach Larsen, Yingqun Nian, Cong Wang, Chunbao Li, Guanghong Zhou

**Affiliations:** 1Key Laboratory of Meat Processing and Quality Control, MOE, Nanjing Agricultural University, Nanjing 210095, China; 2Jiangsu Collaborative Innovation Center of Meat Production and Processing, Quality and Safety Control, Nanjing Agricultural University, Nanjing 210095, China; 3Key Laboratory of Meat Products Processing, MOA, Nanjing Agricultural University, Nanjing 210095, China; 4School of Science, Edith Cowan University, Joondalup, Western Australia 6027, Australia; 5Department of Food Science, Aarhus University, Blichers Allé 20, Tjele 8830, Denmark

**Keywords:** lactose-hydrolyzed milk, β-casein, proteolysis, N-terminal glycated peptides, digestibility

## Abstract

One of the conventional ways to produce lactose-hydrolyzed (LH) milk is via the addition of commercial lactases into heat-treated milk in which lactose is hydrolyzed throughout storage. This post-hydrolysis method can induce proteolysis in milk proteins due to protease impurities remaining in commercial lactase preparations. In this work, the interplay between lactose hydrolysis, proteolysis, and glycation was studied in a model system of purified β-casein (β-CN), lactose, and lactases using peptidomic methods. With a lactase presence, the proteolysis of β-CN was found to be increased during storage. The protease side-activities mainly acted on the hydrophobic C-terminus of β-CN at Ala, Pro, Ile, Phe, Leu, Lys, Gln, and Tyr positions, resulting in the formation of peptides, some of which were N-terminal glycated or potentially bitter. The proteolysis in β-CN incubated with a lactase was shown to act as a kind of “pre-digestion”, thus increasing the subsequent in vitro digestibility of β-CN and drastically changing the peptide profiles of the in vitro digests. This model study provides a better understanding of how the residual proteases in commercial lactase preparations affect the quality and nutritional aspects of β-CN itself and could be related to its behavior in LH milk.

## 1. Introduction

There is an increasing interest in lactose-hydrolyzed (LH) milk, since 70% of the world populations are lactose intolerant, especially in Asian, African, and South American countries [1,2]. In addition, decreasing the content of sugar in food is recommended in many healthy diet programs in China, USA, France, UK, Mexico, Ireland, and South Africa [3]. The production of LH milk usually involves the addition of commercial lactase preparations to hydrolyze the milk lactose into glucose and galactose. The lactase can be added either before (pre-hydrolysis) or after (post-hydrolysis) heat treatment [4]. The post-hydrolyzed LH milk contains active lactase, and the hydrolysis continues throughout the storage of LH milk. The production of high quality post-hydrolyzed LH milk in Northern Europe usually starts with filtration to remove 40% of lactose, which can avoid the excessive sweetness of the product [5]. After that, the product undergoes heat treatment before the addition of a commercial lactase preparation to hydrolyze lactose during storage.

Commercial lactase preparations used in LH milk production are largely isolated from microorganisms by using centrifugation and column chromatography methods, while proteases and other impurities, to varying extents, may still remain [6]. Due to the action of residual proteases in some commercial lactase preparations, recent studies indicated that the proteolysis of milk proteins may occur in post-hydrolyzed LH milk during storage [3,4,7]. The Maillard reaction or so-called “glycation”, a chemical reaction between free amino compounds (e.g., proteins) and carbonyl compounds (e.g., reducing sugars), causes significant changes in the physical and nutritional properties of food. Proteolysis of a protein induced by residual proteases in LH milk can thus generate additional free amino terminals (α-NH_2_) in polypeptides, smaller peptides, or even free amino acids, which may react with glucose and galactose during storage [7,8,9,10,11]. In addition, glucose and galactose, being the hydrolysis products of lactose, are more reactive than lactose in glycation. Therefore, the Maillard reaction rate is greater in post-hydrolyzed LH milk than in conventional milk [10,12]. An elevated Maillard reaction in LH milk contributes to increased levels of volatile and non-volatile glycation compounds, as reported in recent studies [8,9,10]. Therefore, the proteolysis and its interplay with the Maillard reaction in LH milk could partly account for its accelerated quality deterioration and shortened shelf-life compared with conventional ultra heat treated (UHT) milk. Glycated Lys and Arg could be difficultly cleaved during enzymatic digestion, together with the denaturation of protein due to proteolysis, which could affect the digestion behavior of milk protein.

To verify these speculations, a model system of purified β-casein (a main fraction of bovine milk protein) was heat-treated and incubated with two types of commercial lactases (Lactase A or B) that were prepared in different ways by DSM Company (Heerlen, Netherland) in order to study the formation of peptides during storage. Lactase A is identical to lactase B-2, and lactase B to B-1 (less purified), respectively (as taken from a study by Nielsen and colleagues on a milk model [7]). In addition, the in vitro gastrointestinal digestibility of β-casein was further investigated to reveal the potential influence of commercial lactase preparations on the subsequent nutritional aspects of milk proteins. To accelerate the proteolysis process and Maillard reaction, an elevated dosage of commercial lactases (corresponding to at least 10 times the normal dosage used in real production) and a high storage temperature of 38 °C (close to the optimal temperature of many enzymes) were employed. This study will help to increase our understanding of the complex interplay between proteolysis, glycations from the Maillard reaction and digestibility, and, ultimately, how residual protease activity in commercial lactase preparations may affect the quality and nutritional aspects of milk proteins.

## 2. Results and Discussion

### 2.1. Proteolysis of β-casein (β-CN) during Storage 

The proteolysis of β-CN induced by unwanted protease side activities in the studied Lactases A or B was initially assessed by the degree of hydrolysis (DH) and the formation of peptides. Firstly, a fluorescamine assay was applied to monitor the development in the level of free amino groups over storage for up to 2 months with and without the addition of Lactase A or B (Figure 1).

The flourescamine assay measures primary amines and, therefore sums up the net increase in new N-terminals as a result of proteolysis occurring during storage counteracted by the eventual decrease in level of amino groups due to potential glycations of either N-terminals or Lys side chains. It can be seen that the DH value of the control sample without lactase preparation only increased slightly (from 0.24% to 0.73% during 60 days of storage), which could be attributed to the eventual intrinsic protease activity present in the system, not being completely inactivated during the employed heat treatment (95 °C for 5 min). Comparatively, the addition of either Lactases A or B resulted in a more pronounced increase in DH values during 60-day storage, observed to increase (for example) from 0.24% to 5.72% in the sample incubated with Lactase A, and from 0.2% to 8.1% in the Lactase B sample. These results confirm that proteolysis of β-CN induced by proteases activity in the tested commercial lactase preparations occurred during storage, but also that the hydrolysis varies with lactase preparation [4,5,7,8,9].

The peptides generated during storage were identified by LC-ESI-MS/MS Ion Trap. Initially, it was tested and confirmed that no peptides were identifiable before storage in the purified β-CN used as substrate. As shown in Figure 2, only a limited number of peptides was identified in the β-CN control without added lactase, corresponding to its trivial elevation in DH during storage, as revealed in Figure 1, most probably due to the intrinsic milk proteases associated with the used purified β-CN and not inhibited by the heat treatment [13]. In contrast, significantly more peptides were identified in the β-CN incubated with either Lactase A (least) or B, and the number of peptides generated and identified generally showed an increasing trend, which is in line with the higher DH values in Figure 1. Notably, the number of peptides detected in the β-CN incubated with Lactase B decreased from 81 to 76 over the last 30 days of the storage period. This observation indicates that some peptides may be further hydrolyzed into very small peptides or even free amino acids that were beyond the consideration of the proteomics algorithm, which could explain this outcome [14].

Table 1 shows a list of identified representative cleavage sites in β-CN as observed by the peptidomic analyses and assumed to be induced by residual protease activity in the used Lactase A or B preparations, based on the peptides identified by the LC-ESI-MS/MS Ion Trap analysis. No peptides were identified before storage. Notably, all the representative cleavage sites, including Val (170), Lys (176), Ala (177), Ala (189), Phe (190), Leu (191), Tyr (193), Pro (204), Phe (205), Pro (206), Ile (207), and Ile (208), were found to be located near the hydrophobic C-terminus of β-CN. The hydrophobic C-terminus of β-CN was shown earlier to be more vulnerable to cleavage by proteolytic side-activities in both Lactases A or B, corresponding to the earlier observations of Zhao and Rauh [5,13]. Apart from sharing some common cleavage sites for both Lactase A and Lactase B, in Lactase B, some proteotytic side activity further acted before or after the Gln (194), Leu (192), Pro (204) sites. These different purities could be a result of different purification protocols. Additionally, it seems that hydrophobic residues were the main targets for lactase proteolytic side-activities, though in some cases, peptide bonds involving Lys and Ala were cleaved. Turning to LH milk, proteolysis processes during storage may contribute to the sensory properties, either through glycation products related to free or peptide bound amino acids and/or to the formation of bitter peptides [3,4,9,12]. Even through the model system here is different from real LH milk, this study points at a potential contribution of the proteolysis of β-CN as a part of the explanation of changes in sensory attributes during the storage of LH milk.

### 2.2. Formation of Bitter and N-Terminal Glycated Peptide

The proteolysis of β-CN, during storage, generated peptides, some of which were calculated to be potentially bitter. The top twenty identified bitter peptides are illustrated in Table 2. Notably, all of these bitter peptides originate from the hydrophobic C-terminus of β-CN, which corresponded to the representative cleavage sites in Table 1. These bitter peptides could also contribute to the changes in sensory and flavor attributes [8,9,10,11,15].

Glucose and galactose are hydrolysed products of lactose and have a greater activity with free amino groups than its precursor [12,16]. In addition, proteolysis of β-CN induced by residual proteases in commercial lactases creates new α-amino groups at the N-terminus of newly generated peptides. These new α-amino groups could react with glucose or galactose being generated during storage, both in the model system employed here and potentially in post-hydrolysis LH milk. In the present work, six such N-terminally glycated peptides were detected, as indicated in Table 3. Five of these glycated peptides were identified in the β-CN incubated with Lactase B, and two of these peptides were identified in β-CN incubated with Lactase A, in line with the observation of more pronounced proteolysis occurring in Lactase B-treated β-CN during storage compared with Lactase A. The relation between the proteolysis degree and formation of a Maillard reaction-related volatile compound was also reported by Jansson and Jensen [8,9,10].

Notably, N-terminally glycated peptides 1, 3, 4 (from sample incubated with Lactase B) start with Gln(194), which was a representative cleavage site via the proteolysis activity of Lactase B, as shown in Table 1. After lactase addition, the N-termini of these peptides were prone to react with either the glucose or galactose present in the incubated solution, resulting in the formation of N-terminally glycated peptides (peptides 1, 3, and 4). Furthermore, Phe (F 190) or Tyr (Y 193) residues were found to be glycated in two additional peptides, peptides 5 and 6, also representing a major cleavage site, as shown in Table 1. It is possible that such glycated peptides may contribute to the organoleptic properties of LH milk. Flavor compounds such as pyrazinone, furfural, and pyrrole have been reported to be generated during the glycation of peptides, and glycated peptides were important precursors of these compounds [17,18]. Therefore, the formation of these glycated peptides may also contribute to the chemical basis for the increased levels of volatile and non-volatile glycation compounds in post-hydrolysis LH milk, as widely reported in previous work [8,9,10]. These observations points at an interplay between the Maillard reaction and unwanted proteolysis in this system, where the Maillard reaction could be accelerated since proteolysis induced the formation of peptides, which can provide additional amino terminals for glycation by reducing the sugars present. Then, the accelerated Maillard reaction can result in increased levels of volatile and non-volatile glycation compounds during storage.

In addition, glycation has been widely reported to alter the digestibility of protein or peptides [19,20,21,22,23,24]. Therefore, the implications of such N-terminally glycated peptides in relation to digestibility are relevant to consider. Furthermore, the proteolysis of β-CN induced by residual proteases in commercial lactase could break the native structure of β-CN and expose more cleavage sites to digestive proteases during the subsequent in vitro digestion, which may change the digestibility of β-CN to a certain degree. On the other hand, an enhanced Maillard reaction may block trypsin cleavage sites (especially Lys residues) and induce cross-linked structures, which have been widely reported in the literature to reduce protein digestibility [19,20,21,22,23,24]. Accordingly, changes in the digestibility of β-CN after incubation with Lactase A or B are investigated in the following section.

### 2.3. Change in Digestibility

Changes in the digestibility of β-CN, as a result of proteolysis from lactase side activities, as well as an enhanced Maillard reaction due to the presence of glucose and galactose, were studied using the stored β-CN (stored for 30 days at 38 °C with either Lactase A or B) as a substrate for the subsequently added simulated gastric fluid using a static in vitro digestion procedure. This digestion process was analyzed by fluorescamine assay and calculated into DH. It was also studied by SDS-PAGE and the peptidomics of the in vitro digests.

Table 4 shows the values for the DH of in vitro digested samples with and without (control) prior treatment with the commercial preparations of Lactases A or B. In the gastric digestion stage (G), the DH of the control β-CN increased by less than 10%, from 0.4% in G_0_ to 10.0% in G_120_, which is lower than the DH values observed for the β-CN incubated with Lactase A (which increased by 11.0%, from 2.7% to 13.7%) or Lactase B (which increased by 18.2%, from 4.2% to 22.4%) after the 120 min of digestion with pepsin at 37 °C. These results indicate a more pronounced proteolysis to occur during the gastric digestion of β-CN “pre-incubated” with commercial lactase preparations in a so-called “post-hydrolysis” step. The proteolysis induced by residual proteases in the commercial lactase preparations may have broken up the native structure of β-CN, especially in the β-CN sample incubated with Lactase B (less purified) and also shown by the SDS-PAGE analysis (Figure 3C, lane 0). Hypothetically, more residues inside the protein may be exposed and become more accessible to pepsin hydrolysis during the simulated gastric digestion, thereby promoting the gastric digestion of the β-CN incubated with the commercial lactase preparations. After 120 min of gastric digestion, intact β-CN was still be observed to be present in the control sample (Figure 3A, lane 5), in contrast to the situations with lactase additions at the same time point (Figure 3B, lane 5, Figure 3C, lane 5). According to the DH values and SDS-PAGE results, the gastric digestibility of β-CN was largely elevated in the β-CN pre-incubated with commercial lactases, due to the pre-proteolysis during the initial post-hydrolysis storage.

For the intestinal digestion stage, after 120 min of digestion, the DH of control β-CN increased by almost 49% from G_120_ to GI_120_, while 55% and 52% were measured for the β-CN incubated with either Lactase A or B, respectively. This showed the increased digestibility of β-CN also pre-treated with lactase in the intestinal phase, though glycations may have limited the differences. This, however, remains to be further studied. Glycations have been widely reported to block the cleavage sites (especially Lys residues) for trypsin, thereby reducing the digestibility of lactase-treated milk proteins (i.e., galactose and glucose-containing systems) [19,20,21,22,23,24]. The presumably higher extent of the Maillard reaction in β-CN incubated with Lactase B, as verified in the present section (Table 3), may contribute to its smaller increase in DH (52%) from G_120_ to GI_120_, compared with the value for β-CN incubated with Lactase A (55%) in the intestinal digestion stage [6,7,8]. For the entire gastrointestinal digestion, the proteolysis during the so-called post-hydrolysis pre-storage with lactases enhanced the digestibility of β-CN during the subsequent gastrointestinal digestion, enhancing DH from 59% (control β-CN) to 69% (β-CN + Lactase A) and 75% (β-CN + Lactase B). Proteolysis during the “post-hydrolysis” storage of β-CN was thus shown to act as a “pre-digestion” role and largely elevated the digestibility of the tested protein.

Differences in the peptide compositions of the in vitro gastric (G) and gastrointestinal (GI) digests were further investigated by peptidomics, and the results are shown in Table 5 and Figure 4. After gastric digestion, 14 and 32 new peptides were found in the gastric digests of β-CN incubated with either Lactase A or B, respectively, whereas 16 and 15 peptides that were identified in the control sample were absent in the lactase-treated samples. These results demonstrate the significant changes in the peptide compositions of the gastric digests of β-CN, as induced by the proteolytic side activities present in the lactase preparations. Notably, most of these new gastric peptides were found to be located near the β-CN C-terminal, corresponding to the more pronounced proteolysis in this region during post-hydrolysis storage at 38 °C [7]. This result directly indicated the influence of post-hydrolysis induced proteolysis of β-CN due to lactase side activities on its subsequent gastric digestion behavior.

Compared with the peptide composition after gastric digestion, smaller changes in the peptide composition of gastrointestinal digests were found. Eight and 19 new peptides that were found in the gastrointestinal digests of β-CN incubated with Lactase A or B, respectively, whereas 6 and 10 peptides in the control sample were absent in the lactase-treated samples. The abundant proteases in pancreatin could have efficiently hydrolyzed the gastric digests of β-CN and thus could minimize the “pre-digestion” influence of proteolysis during treatment with the lactases on the intestinal digestion of gastric-digested β-CN. No N-terminally glycated peptides were found in neither gastric nor gastrointestinal digests of β-CN, possibly because these N-terminally glycated peptides (Table 3) had been cleaved into small peptides that were beyond the consideration of the proteomic algorithm.

Taken together, the results obtained from the DH assessment, the SDS-PAGE analysis and peptidomics clearly indicate the changes in the digestion behavior of the lactase treated β-CN in the used model system, especially in the gastric stage. Proteolysis played a “pre-digestion” role during the post-hydrolysis step and largely elevated the digestibility of β-CN. In addition, the pronounced Maillard reaction in the β-CN incubated with Lactase B may have had an opposed effect by reducing protein digestibility in the intestinal digestion stage. In the study of Nielsen, flocculation and the formation of crosslinked structures that may affect the protein digestibility were found prone to occur in LH milk [7]. These results indicate that the digestibility of milk proteins in post-hydrolysis LH milk can be largely different from that of conventional milk. However, the nutritional significance of these changes requires more work, and more studies on the contributions of the interplay between the mechanisms of initial Maillard reactions and proteolysis in real LH systems are warranted.

## 3. Materials and Methods

### 3.1. Materials

β-Casein (β-CN) was purified from the milk obtained from the research herd at Aarhus University (Tjele, Denmark) using the cooling and centrifugation methods published earlier [5]. Commercial lactase preparations A and B were obtained from the DSM Company (Heerlen, Netherland). Lactase A is identical to Lactase B-2, and Lactase B to B-1, respectively, as studied earlier by Nielsen [7]. Fluorescamine (≥98%, CAS No.38183-12-9), leucine (≥98%, CAS No.61-90-6), pepsin (from porcine, ≥250 unit/mg, CAS No.9001-75-6), and pancreatin (from porcine, 8 × USP, CAS NO. 8049-47-6) were obtained from Sigma–Aldrich (Steinheim, Germany). The activity of pepsin and pancreatin was evaluated according to the method published earlier [25].

### 3.2. Storage Experiment

β-CN was dissolved to 10 mg/mL in 0.025 M sodium phosphate buffer (pH 6.5, 0.05% sodium azide) and heated at 95 °C for 5 min in a water bath to inactivate the potential indigenous milk enzymes in the β-CN preparation. From this β-CN stock solution, samples containing an additional 4.8% (*w*/*v*) lactose were prepared as a model of conventional milk (control sample) [5]. Furthermore, β-CN samples containing 3.0% (*w*/*v*) lactose (corresponding to a situation where 40% of lactose was removed by filtration, as is normal practice in Northern European countries to avoid excessive sweetness) and 0.09% (*v*/*v*) of either Lactase A (β-CN + Lactase A) or B (β-CN + Lactase A), were prepared as models of the post-hydrolysis process in the production of LH milk [5]. Each protein sample was aliquoted into 5 mL sealed bottles to reach a final concentration of 5 mg/mL and a final volume of 2.5 mL, and stored at 38 °C for 0, 10, 20, 30, 40, or 60 days. The experiment was carried out in duplicate. Samples were collected and stored at −80 °C before being analysed. It is noted that a higher temperature than the usual milk-storage-temperature, as well as a higher level of lactase (whereas in the production of LH milk, 0.004%–0.010% is normally employed) was used to accelerate the proteolysis and amplify the influence of proteolysis on protein digestibility.

### 3.3. In Vitro Digestion

An in vitro static digestion system was applied [5,25]. For the gastric digestion step, 2 mL of the β-CN post-hydrolysis solution sample was mixed with 4 mL of simulated gastric fluid containing 35 mM hydrochloric acid, 1.8 mM mono potassium phosphate, 50 mM sodium bicarbonate, 100 mM sodium chloride, 0.2 mM magnesium chloride, and 1 mM ammonium carbonate, pH 3.0. Then, a pepsin solution (10 mg/mL) was added to reach a final activity of 500 units/mL, and the mixture was incubated at 37 °C for 120 min. After that, 3 mL of simulated intestinal fluid (SIF) containing 6.8 mM potassium chloride, 0.8 mM mono potassium phosphate, 85 mM sodium bicarbonate, 38.4 mM sodium chloride, 0.66 mM magnesium chloride, and 8.4 mM HCl was added to elevate the pH to 7 and inactivate pepsin. At each set time points of 1, 5, 15, 60, and 120 min, 200 μL of the digested sample was withdrawn and mixed with 150 μL of SIF to inactivate the pepsin and prime for intestinal digestion conditions. For the intestinal digestion, pancreatin was added after the addition of SIF to reach a final activity of 5 p-toluene-sulfonyl-L-arginine methyl ester (TEME) units/mL. This mixture was incubated at 37 °C for 120 min, and the reaction was stopped by heating at 100 °C for 3 min. At each time point of 1, 5, 15, 60, and 120 min, a 300 μL aliquot was obtained, heated at 100 °C for 3 min to inactivate the pancreatin, and stored at −20 °C prior to further analyses.

### 3.4. SDS-PAGE

Digests were analyzed by reducing SDS-PAGE on a Mini Protean II system (Bio-Rad Laboratories, Richmond, CA, USA) using precast gels (NovexTM, 10–20%, Thermo Fisher, San Diego, CA, USA) [5]. The digested samples were diluted 5-fold with a sample buffer containing 0.2 M Tris, 0.2% SDS, 4% glycerol, 0.05% Coomassie G250, and 0.02 mM dithioerythritol, and heated at 90 °C for 5 min. Aliquots of the diluted samples (20 μL) were loaded onto a precast gel and run at 125 V before staining with Coomassie G250.

### 3.5. DH

The DH of the digests was detected based on a fluorescamine method [26]. The digested sample (75 μL) was mixed with 75 μL of 24% trichloroacetic acid and precipitated on ice for 30 min. The solution was then centrifuged at 13,000 rpm for 20 min at 4 °C. After that, 30 μL standard (L-leucine, 0.05–3 mM) or sample supernatant was withdrawn and mixed with 900 μL sodium tetraborate (0.1 M, pH 8.0), and 300 μL of fluorescamine acetone solution (0.2 mg/mL) was added. Fluorescence was measured using excitation and emission wavelengths of 390 and 480 nm, respectively. The DH was calculated as follows:$$\mathrm{DH}\;=\;\frac{\left[-{\mathrm{NH}}_{2}\;\left(h\right)\right]-\left[-{\mathrm{NH}}_{2}\;\left(0\right)\right]}{\left[-{\mathrm{NH}}_{2}\;\left(\infty \right)]-[-{\mathrm{NH}}_{2}\;\left(0\right)\right]}.$$

The [-NH_2_] indicates the concentration of the primary amines in the hydrolyzed (h) or unhydrolyzed (0) sample, and [-NH_2_ (∞)] indicates the theoretical maximal primary amine concentration and was calculated as follows:$$\left[-{\mathrm{NH}}_{2}\left(\infty \right)\right]=\;\frac{\left[1+f\;\left(\mathrm{Lys}\right)\right]C}{\mathrm{MW}\left(\mathrm{AA}\right)}$$
where f (Lys) indicates the relative content of lysine in β-CN, C indicates protein concentration, and MW (AA) indicates the mean molecular weight of amino acids in the protein.

### 3.6. LC-ESI-MS/MS Ion Trap

Peptides that were generated during storage or by in vitro digestion were identified by LC-ESI-MS/MS Ion Trap [27]. Samples were filtrated through 10 kDa cut-off spin filters at 14,000× *g* (4 °C) for 15 min. Then, the filtered sample (10 µL) was loaded onto an Aeris Peptide C18 column (250 × 2.1 mm, 3.6 μm, Phenomenex, Torrance, CA, USA) connected to an Agilent LC 1200 and an HCT Ultra Ion Trap (Bruker Daltonics, DE, Frederikssund, Denmark). A gradient elution was applied by using 0.1% formic acid (solvent A) and was 90% acetonitrile in 0.1% formic acid (solvent B) as follows: 0–5 min, 2% B; 5–85 min, 40% B; 85–105 min, 80% B; for 105–110 min, 2% B. The MS and MS/MS masses were scanned in the range of 300–1800 *m*/*z* and 150–1800 *m*/*z*, respectively (small peptides (smaller than 5 amino acids) are not considered). The obtained tandem mass data were searched using Mascot v2.4 (Matrix Science, London, UK) against a custom-made database, as described earlier [5,28]. The following parameters were applied: protease, unspecified; mass tolerance for the precursor iron was set as 15 ppm and 0.6 Da for the product irons; variable modification: phosphorylation for Ser and hexose for Lys. Each sample was examined twice, and only peptides identified as significant (Mascot score > 26, expected *p* < 0.05) in both replicates are displayed in this work.

The hydrophobic index, determined as the Q-value, was calculated for the identified peptides to predict their bitterness potential using the algorithm proposed by Ney [29].

### 3.7. Statistics Analysis

An analysis of variance (ANOVA) of the DH values was performed using the SPSS 20 software (SPSS inc., Chicago, IL, USA). Statistical differences were determined by a one-way analysis of variance with Duncan’s post hoc test. Differences were significant when *p* value < 0.05.

## 4. Conclusions

Proteolytic activities from residual protease side activities in commercial lactases were demonstrated in a model system of β-CN and lactose at conditions mimicking the accelerated post-hydrolysis storage of LH milk. Bitter and N-terminally glycated peptides derived from the hydrophobic C-terminus of β-CN were detected. This points to a synergistic relationship between proteolysis and the initial Maillard reaction processes occurring during storage for induced quality changes. Additionally, the in vitro digestibility of β-CN was largely affected by the proteolytic activities of commercial lactases during storage, resulting in elevated DH and changes in the peptide composition of digests compared to the digests of the control sample. This study helps to understand how residual protease activity in commercial lactase preparations can affect the quality and nutritional features of β-CN and potentially the interplay of the processes in LH milk.

## Figures and Tables

**Figure 1 molecules-24-02876-f001:**
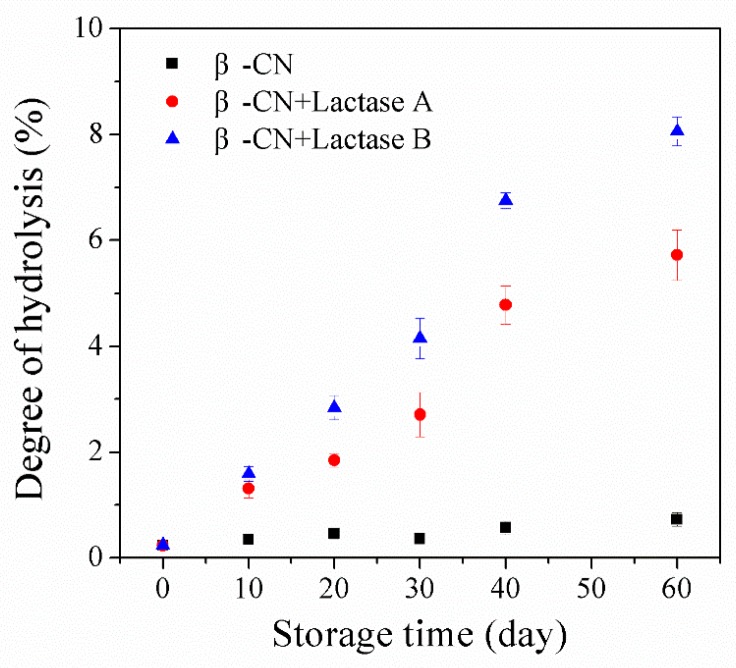
Increase in the degree of hydrolysis (DH) as assessed by the fluorescamine assay in ccontrol sample and sample with commercial lactase preparations A or B (0.09% *v*/*v*). The data are the average of triplicates ± standard deviations.

**Figure 2 molecules-24-02876-f002:**
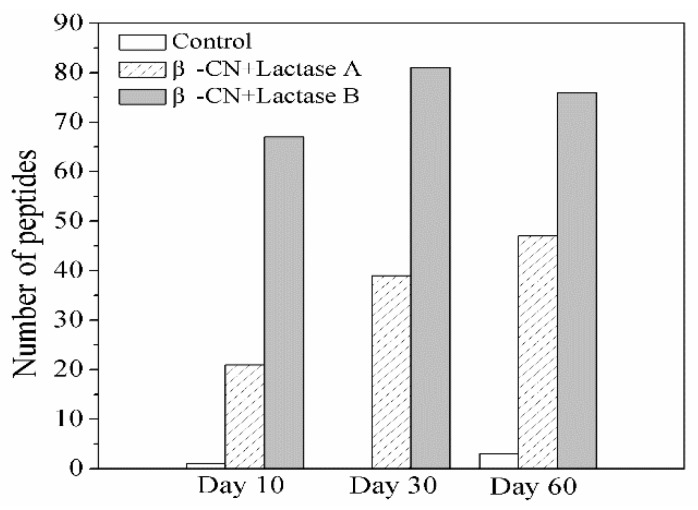
Change in the type of identified peptides from β-CN by LC-ESI-MS/MS in the control sample and sample incubated with commercial Lactases A or B after 10, 30, and 60 days of storage at 38 °C.

**Figure 3 molecules-24-02876-f003:**
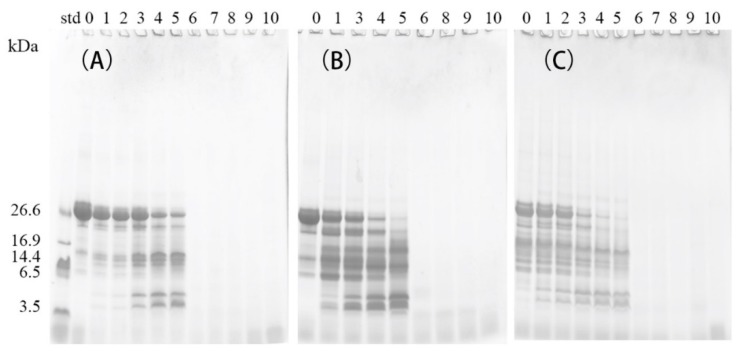
SDS-PAGE analysis of digests using the simulated gastric fluid of the control β-CN without added Lactase (**A**), and β-CN incubated with either Lactase A (**B**) or Lactase B (**C**). Lane 0 indicates undigested samples; lanes 1, 2, 3, 4, and 5 indicate samples digested for 1, 3, 15, 60, or 120 min in the gastric stage, respectively; lanes 6, 7, 8, 9, and 10 indicate samples digested for 120 min in the gastric stage, followed by 1, 3, 15, 60, or 120 min digestion in the intestinal stage, respectively.

**Figure 4 molecules-24-02876-f004:**
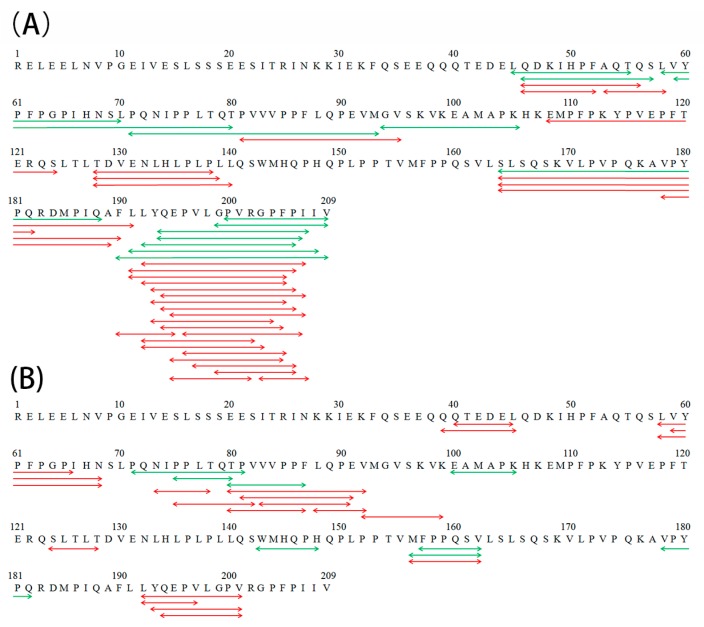
New peptides in the gastric (**A**) and gastrointestinal (**B**) digests of the β-CN incubated with either Lactase A (green line) or B (red line), as identified by peptidomics.

**Table 1 molecules-24-02876-t001:** Representative cleavage sites of β-casein (β-CN) by residual proteases in commercial Lactase A or B. The cleavage sites were selected on the basis of their frequencies identified at the N- or C-terminal ends of the identified peptides. Residues that were repeatedly (more than five times) identified to locate at the N- or C-terminal ends of peptides are illustrated.

Sample	Representative Cleavage Sites
β-CN + Lactase A (day 10)	F (190), I (207), I (208)
β-CN + Lactase A (day 30)	A (177), P (206), K (176), I (207), F (190), I (208), L (191), F (205), Y (193)
β-CN + Lactase A (day 60)	A (177), F (205), P (206), K (176), I (207), F (190), I (207), I (208), L (191), E (195),
β-CN + Lactase B (day 10)	F (190), V (170), P (206), F (205), Y (193), A (189), Q (194), L (191), L (192)
β-CN + Lactase B (day 30)	F (190), V (170), P (206), F (205), Y (193), A (189), Q (194), L (192), A (177), P (204)
β-CN + Lactase B (day 10)	I (207), F (190), P (206), I (208), Y (193), A (189), Q (194), L (192), A (177), S (164), I (207)

**Table 2 molecules-24-02876-t002:** Representative bitter peptides from β-CN incubated with Lactase A or B. The peptides were selected according to their Q value in a descending order.

No.	Peptides	Identified Mass (Da)	Q Value	Origin
1	G(203)-V(209)	642.37	2305	β-CN + Lactase B
2	G(199)-V(209)	1150.69	1869	β-CN + Lactase B
3	E(195)-I(208)	1489.87	1868	β-CN + Lactase B
4	E(195)-I(209)	1588.93	1856	β-CN + Lactase B
5	L(171)-I(187)	1948.06	1840	β-CN + Lactase A
6	Y(193)-I(208)	1780.99	1808	β-CN + Lactase B
7	Y(193)-V(209)	1880.06	1801	β-CN + Lactase A or B
8	F(190)-P(206)	1928.06	1792	β-CN + Lactase A
9	V(178)-I(187)	1214.61	1786	β-CN + Lactase A
10	G(199)-I(207)	938.53	1767	β-CN + Lactase B
11	V(170)-Q(182)	1434.82	1759	β-CN + Lactase B
12	F(190)-F(205)	1831.00	1741	β-CN + Lactase B
13	L(191)-P(206)	1780.99	1739	β-CN + Lactase A or B
14	V(178)-R(183)	758.41	1738	β-CN + Lactase B
15	A(177)-V(209)	3720.03	1737	β-CN + Lactase B
16	Y(193)-I(207)	1667.90	1734	β-CN + Lactase B
17	Q(194)-V(209)	1716.99	1730	β-CN + Lactase B
18	D(184)-I(207)	2696.44	1729	β-CN + Lactase B
19	V(170)-I(187)	2047.13	1724	β-CN + Lactase A
20	L(171)-P(186)	1834.98	1704	β-CN + Lactase A

**Table 3 molecules-24-02876-t003:** N-terminally glycated (by glucose or galactose) peptides identified by an LC-ESI-MS/MS Ion Trap after 30 days of storage at 38 °C.

No.	N-Terminally Glycated Peptides	Identified Mass (Da)	Theoretical Mass (Da)	Origin
1	*Q*(194)EPVLGPVRGPFPI	1667.90	1668.15	β-CN + Lactase B
2	*Q*(188)AFLLYQEPVLGPVRGP	2046.09	2046.37	β-CN + Lactase A
3	*Q*(194)EPVLGPVRG	1212.59	1212.64	β-CN + Lactase B
4	*Q*(194)EPVLGPVRGP	1309.83	1309.69	β-CN + Lactase B
5	*F*(190)LLYQEPVLGP	1536.60	1535.81	β-CN + Lactase A or B
6	*Y*(193)QEPVLGPVRGPFP	1716.35	1716.87	β-CN + Lactase B

**Table 4 molecules-24-02876-t004:** Degree of hydrolysis (DH) of β-CN (stored for 30 days at 38 °C) during in vitro gastrointestinal digestion in simulated gastric fluid. G_0_ indicates DH before digestion; G_15_ and G_120_ after 15 and 120 min of gastric digestion, respectively; GI_15_ and GI_120_ indicate the DH of β-CN after 120 min of gastric digestion, followed by 15 or 120 min of intestinal digestions, respectively. The data are the average of triplicates ± standard deviations. Different letters (a, b, and c) within columns denote levels of significant difference (*p* < 0.05).

Sample (Day 30)	DH (%)				
	G_0_	G_15_	G_120_	GI_15_	GI_120_
β-CN (control)	0.36 ± 0.06^a^	4.37 ± 0.37^a^	9.97 ± 0.87^a^	44.28 ± 3.63^a^	58.91 ± 2.81^a^
β-CN+Lactase A	2.71 ± 0.42^b^	10.72 ± 1.83^b^	13.74 ± 2.10^b^	60.82 ± 4.17^b^	69.19 ± 4.37^b^
β-CN+Lactase B	4.15±0.04^c^	16.57 ± 2.19^c^	22.41 ± 3.04^c^	67.53 ± 5.82^b^	74.85 ± 5.80^c^

**Table 5 molecules-24-02876-t005:** Number of peptides in digests of the control β-CN and sample incubated with Lactase A or B after gastric (G) and gastrointestinal (G + I) digestion, as identified by LC-ESI-MS/MS Ion Trap. Missed peptides indicate peptides identified only in the control sample digests, and new peptides indicate those identified only in samples with added Lactase A or B.

Samples	Peptides after G Digestion			Peptides after G+I Digestion		
	Total	Missed	New	Total	Missed	New
β-CN (control)	63	_	_	53	_	_
β-CN + Lactase A	61	16	14	55	6	8
β-CN + Lactase B	80	15	32	62	10	19

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
