# Peer review of "Interplay between Residual Protease Activity in Commercial Lactases and the Subsequent Digestibility of β-Casein in a Model System"

_molecules, 2019, doi:10.3390/molecules24162876_

Round 1

Reviewer 1 Report

The manuscript entitled “Residual proteases in commercial lactase preparations hydrolyze
β-casein during storage: Formation of N-terminal glycated peptides during
storage and change in digestibility” (Manuscript ID: molecules-498291), investigates protease activity of commercial lactase preparations. The manuscript includes new information about influence lactase activity on β-casein of milk. The study has some merits, there is  number of deficiencies that need to be addressed before it can be accepted for publication.

The major issues of this paper are as follows:

Why did you assumed that lactase preparations      have residual proteases? Was it checked or did you have Specification      Sheet? Maybe lactase preparations had also protease activity?

Statistical method is missing.

Why did you use only β-casein as a material      instead milk?

Detailed comments:

Line 26: If you use only one protein fraction of milk the conclusion is too generally.

Line 45: Why Millard reaction? Milk is not storage at high temperature.

Line 61: Are you sure that lactase preparations have residual proteases?

Line 64: It should be A or B.

Figure 2: Why did you determined number of peptides only for 10, 30 and 60 days? And the other samples? The sample before storage and statistical method are missing.

Why number of peptides in control decreased after 10 days of storage and then next increased? Should be clarified and discussed.

Table 1. Why did you chose this sample? Control sample is missing.

Line 193: “proteolysis during storage on the intestinal digestion” it is not clear.

Materials and Methods

Line 209: The abbreviation should be clarified.

Line 223: Why did you storage the samples at 38 °C?

Line 223: It should be: Samples were collected and storage at -80 °C.

Line 226: The reference should be presented according to Author Guide.

Conclusion should be rewritten because the material of study was not milk.

Author Response

Response to reviewer’s comments

The manuscript entitled “Residual proteases in commercial lactase preparations hydrolyze β-casein during storage: Formation of N-terminal glycated peptides during storage and change in digestibility” (Manuscript ID: molecules-498291), investigates protease activity of commercial lactase preparations. The manuscript includes new information about influence lactase activity on β-casein of milk. The study has some merits, there is number of deficiencies that need to be addressed before it can be accepted for publication.

Response: Thank you for your comments and suggestions, and they are helpful for the improvement of this manuscript. We have made a thorough revision of the entire manuscript based on the comments. A new table (Table 2) was provided, and more data was added to Table 1. A new title has been provided, and sample information as well as statistical methods description have been added in the revised manuscript. In addition, the language have been carefully revised. Moreover, the format of the references was adjusted according to the requirement of Molecules.

Why did you assumed that lactase preparations have residual proteases? Was it checked, or did you have Specification Sheet? Maybe lactase preparations had also protease activity?

Response: The commercial lactases were prepared from microorganism by using centrifugation and chromatography methods, whereas some impurities were very difficult to be separated from the galactosidase which is the active fraction in commercial lactases to hydrolyze lactose. The SDS-PAGE images of the two commercial lactases preparations used (A and B) are shown in Figure S1, indicating that each one consists of galactosidase and some impurities. Galactosidase acts at the glycosidic bond of lactose, not at the peptide bonds of β-CN. Therefore, the proteolysis should originate from some proteases, present as side activity due to impurities of the commercial lactases. It has been published earlier, that these side activities are present, and can hydrolyse the milk proteins to varying degree, depending on conditions and enzyme type (references 3, 5 and 8). Furthermore, the two enzymes used here, A and B, are equivalent to earlier used and published enzymes (Lactase A is identical to lactase B-2, and lactase B to B-1 (less purified), respectively, from the study of Nielsen and colleagues in a milk model, Proteolytic side activity of lactase preparations, Int Dairy J., 2018, 78, 158-168). This important information has been added to materials and methods section, and this reference included.

Figure S1. Commercial lactase preparation A and B (unpublished)

Statistical method is missing.

Response: The statistical analysis method was provided in the revised manuscript (line 353-356), please check.

Why did you use only β-casein as a material instead milk?

Response: we also conducted some work using milk before starting this work, and some digestion results were obtained (Table 1). After that, we decided to simplify the model by using purified protein, so that we can in depth investigate the relationship between proteolysis during storage and digestive changes of milk protein. Then we chose β-CN which is a main fraction of milk protein.

Table S1. Liberation of free amino group during the gastrointestinal digestion of stored milk with and without addition of a commercial lactase preparation (unpublished).

Detailed comments:

Line 26: If you use only one protein fraction of milk the conclusion is too generally.

Response: We agree that we should be cautious when expanding the conclusion from β-CN to milk. Therefore, conclusion and relative parts was revised extensively in the new manuscript, line 27, 125-130, 367 for examples.

Line 45: Why Millard reaction? Milk is not storage at high temperature.

Response: Maillard reaction also occur in room temperature in milk sample, especially in LH milk system. Figure S2 illustrate the photograph of milk sample after 60 days of storage at 38 °C with and without addition of commercial lactase preparation. The “brown” color was clearly shown in milk incubated with a commercial lactase, which is the typical signal of the occurrence of Maillard reaction. In addition, many previous results (reference 4, 8-11) indicated that the Maillard reaction occur more pronouncedly in LH milk than conventional milk due to the proteolysis (by residual proteases) of milk protein and hydrolysis of lactose (into glucose and galactose). Maillard reaction can reduce protein digestibility by blocking Lys and Arg residues (reference 19-24). That’s why Maillard reaction was considered in this study.

Figure S2. photograph of milk sample after 60 days of storage at 38 °C with (+Lactase A, +Lactase B) and without (control) addition of commercial lactase preparation (unpublished result).

Line 61: Are you sure that lactase preparations have residual proteases?

Response: Please refer to Figure 1 and previous response.

Line 64: It should be A or B.

Response: Yes, it should be A or B. This expression was revised throughout this manuscript.

Figure 2: Why did you determined number of peptides only for 10, 30 and 60 days? And the other samples? The sample before storage and statistical method are missing.

Response: No peptides was identified before storage (line 115-116). We selected sample that were collected at these time points to study the peptides formation at the initial, middle and final stage of storage.

Why number of peptides in control decreased after 10 days of storage and then next increased? Should be clarified and discussed.

Response: 1, 0 and 3 peptides were identified in control sample after 10, 30 and 60 days of storage, this is just the result we obtained. These results indicate that few peptides were identified significantly in control β-CN, which correspond to DH results in Figure 1. Instead, we interpreted the decreased types of peptides in sample incubated with Lactase B for 60 days compared with that stored for 30 days (line 106-108).

Table 1. Why did you choose this sample? Control sample is missing.

Response: The information of sample incubated for 10 and 60 days were added to Table 1. Residues that were repeatedly (more than five times) identified to locate at the N- or C-terminal ends of peptides were illustrated (line 116-120). Only 0-3 peptides were identified in control sample, no residues were found repeatedly more than five times. That is why control sample was not shown. Instead, more data of lactase-treated samples was added to Table 1.

Line 193: “proteolysis during storage on the intestinal digestion” it is not clear.

Response: We rewrote this sentence in the revised manuscript to make it clear, please check this sentence in line 253-265.

Materials and Methods

Line 209: The abbreviation should be clarified.

Response: The abbreviation was specified in the revised manuscript. 

Line 223: Why did you storage the samples at 38 °C?

Response: This is model study. The storage temperature was set at 38 °C to accelerate the proteolysis and Maillard reaction and amplify the influence of proteolysis and Maillard reaction on protein digestibility, since many proteases act efficiently at this temperature (line 71).

Line 223: It should be: Samples were collected and storage at -80 °C.

Response: Yes, it should be written in this way (line 295-296).

Line 226: The reference should be presented according to Author Guide.

Response: The references were checked and revised according to Author Guide in the revised manuscript, please check.

Conclusion should be rewritten because the material of study was not milk.

Response: The conclusion section was revised to confine the conclusion in β-CN.

Reviewer 2 Report

General comments:

The manuscript is very difficult to read due to severe language problems. I strongly suggest the authors ask for English editing service before the second round review. The title is confusing and needs to be re-write. The hypothesis of study is vague and many details are missing. For specific terminologies, brief descriptions are necessary to help general readers understand. Sample size information, replicates (inter- and inter- CVs), statistics need to be added. Discussion part is not thorough, I do not see deeply interpretation of results. 

Specific comments:

Line (L) 45: Describe ‘Maillard reaction’.

L62: ‘Results and discussion’ section needs more detailed data interpretation. 

L64: What is the rationale to used lactase A and B? What are the differences? The information needs to be added to ‘Introduction’.

L66: What is the sample size?

L82: What do you mean number of peptides? It is not convincing to me. Quantitation or relative abundances of peptides need to be done. 

L83-87: Give background information of different cleavage sites. What do they mean? 

L96: Table 1 needs more details. Hard to understand. 

L110: Table 2. How did you determine glycated sequence based on mass? Identified mass is different from theoretic mass. How did you do annotation? 

L131: Section 2.2 appeared twice. Change 2.2 to 2.3. 

Author Response

Response to reviewer

The manuscript is very difficult to read due to severe language problems. I strongly suggest the authors ask for English editing service before the second-round review. The title is confusing and needs to be re-write. The hypothesis of study is vague and many details are missing. For specific terminologies, brief descriptions are necessary to help general readers understand. Sample size information, replicates (inter- and inter- CVs), statistics need to be added. Discussion part is not thorough, I do not see deeply interpretation of results.

Response: Thank you for your comments and suggestions, and they are helpful for the improvement of this manuscript. We have made a thorough revision of the entire manuscript based on the comments. A new table (Table 2) was provided, and more data was added to Table 1. A new title has been provided, and sample information as well as statistical methods description have been added in the revised manuscript. In addition, the language have been carefully revised. Moreover, the format of the references was adjusted according to the requirement of Molecules.

Specific comments:

Line (L) 45: Describe ‘Maillard reaction’.

Response: The “Maillard reaction” was described in the revised manuscript (line 47-50).

L62: ‘Results and discussion’ section needs more detailed data interpretation.

Response: More information had been added throughout the “Results and discussion” section, please check.

L64: What is the rationale to used lactase A and B? What are the differences? The information needs to be added to ‘Introduction’.

Response: These commercial lactases were prepared in different way by the company. We added some information in the revised manuscript (line 65-66). The SDS-PAGE images of the two commercial lactases preparations were shown in Figure S1, indicating that each one consists of galactosidase and some impurities.

Figure S1. Commercial lactase preparation A and B

L66: What is the sample size?

Response: Each sample (2.5 mL) for each time point was stored in an individual 5 mL sealed bottle, each bottle was sealed until it collected at set time. Each sample was prepared in duplicates, as shown in Figure S2. This information was added in the revised manuscript (line 293-295).

Figure S2. The stored β-CN samples with and without a commercial lactase.

L82: What do you mean number of peptides? It is not convincing to me. Quantitation or relative abundances of peptides need to be done.

Response: It was the types of identified peptides. The caption of Figure 2 was revised as “Representative cleavage sites of β-CN by residual proteases in commercial Lactase A or B. The cleavage sites were selected on the basis of their frequencies identified at the N- or C-terminal ends of the identified peptides. Residues that were repeatedly (more than five times) identified to locate at the N- or C-terminal ends of peptides were illustrated”. 

L83-87: Give background information of different cleavage sites. What do they mean?

Response: the background information of different cleavage sites was provided in the revised manuscript (line 113-115, line 131-134).

L96: Table 1 needs more details. Hard to understand.

Response: The caption of Table 1 was rewritten as: Representative cleavage sites of β-CN by residual proteases in commercial Lactase A or B. The cleavage sites were selected on the basis of their frequencies identified at the N- or C-terminal ends of the identified peptides. Residues that were repeatedly (more than five times) identified to locate at the N- or C-terminal ends of peptides were illustrated.

L110: Table 2. How did you determine glycated sequence based on mass? Identified mass is different from theoretic mass. How did you do annotation?

Response: Information regarding the determination of glycated sequence was added in the method section, please refer to line 346-349 in the revised manuscript: variable modification: phosphorylation for Ser and hexose for Lys.

L131: Section 2.2 appeared twice. Change 2.2 to 2.3.

Response: Yes, it should be 2.3.

Round 2

Reviewer 1 Report

All comments have been corrected. The manuscript can be accepted in present form.